# The Synthesis and Utility of Metal-Nitrosophenolato Compounds—Highlighting the Baudisch Reaction

**DOI:** 10.3390/molecules24224018

**Published:** 2019-11-06

**Authors:** Alexander J. Nicholls, Thomas Barber, Ian R. Baxendale

**Affiliations:** Department of Chemistry, University of Durham, South Road, Durham DH1 3LE, UK; alexander.j.nicholls@durham.ac.uk (A.J.N.); Thomas.Barber@nottingham.ac.uk (T.B.)

**Keywords:** *C*-nitrosation, copper complexes, Baudisch reaction, *ortho*-nitrosophenols

## Abstract

The syntheses of the title compounds demonstrate a privileged introduction of a nitroso (and a hydroxyl via the Baudisch reaction) group to an aromatic ring. These complexes first appeared in the literature as early as 1939, and a range of applications has subsequently been published. However, optimisations of the preparative sequences were not considered, and as such, the reactions have seldom been utilised in recent years; indeed, there remains confusion in the literature as to how such complexes form. In this review, we aim to demystify the misunderstanding surrounding these remarkable complexes and consider their renewed application in the 21st century.

## 1. Introduction to Metal-Nitrosophenolato Complexes

Metal-nitrosophenolato complexes consist of a metal ion (most commonly copper(II)) flanked by typically two or more 2-nitrosophenolate ligands. The structures of the nitrosophenols (both as ligands and free molecules) are known to resonate between the nitrosophenol and the quinone-monoxime forms [1,2,3]. Although the monoxime is considered to dominate, most literature representations display the nitrosophenol form, often for clarity and simplicity. Interestingly, the nitrosophenolato complexes are often highly coloured and reported to possess only limited solubility in most common organic solvents and water. Historically, the complexes have featured in a wide range of publications most often-based upon their colourimetric properties, although in recent years, both the metal complexes and their 2-nitrosophenol ligands have received little attention, becoming an almost forgotten class of molecules. However, one compound that has gained current commercial applicability [4] is an iron *tris*(1-nitroso-2-naphtholato), a dye known as ‘pigment green 8’ (Figure 1) [5,6,7]. This vivid green species has found widespread application in concrete, textiles, paint and rubber colouration.

One of the reasons why this area of chemistry has been somewhat overlooked in recent years is most likely due to the background literature being spread widely, and often thinly, across many subdomains of Organic, Inorganic, Analytical and Materials Chemistry. However, many positive factors, which we will explore subsequently, such as the affordability and availability of starting materials, the mild reaction conditions, lack of organic solvent-requirements and applicability to naturally occurring aromatic feedstocks make this topic relevant to the chemical research community today more than ever before.

### 1.1. Metal-Nitrosophenolato Complexes in Nature

Remarkably, substituted iron *tris*(nitrosophenolato) complexes are naturally biosynthesised by particular Streptomycete bacteria as specialised antibiotics [8,9,10,11]. Ferroverdin (Figure 2) was the first confirmed natural product to be isolated [8], although this was rapidly followed by the related Viridomycin A, Actinoviridin and Viridomycin E [12]. These complexes were found to target certain bacteria by disrupting their cell membranes; however, the free ligands were determined to be more effective than the corresponding iron complexes [13]. Interestingly, despite the presence of three negatively charged ligands, the complexes all exist in the Fe(II) oxidation state, with the overall negative charge counterbalanced with a sodium or equivalent cation [9,14]. It should, however, be acknowledged that this general characterisation is not universally accepted [12], and is complicated by the fact that the compounds are redox active and that, although the complexes are largely inert [13], ligands can be added/removed under a range of mild conditions [14]. Interestingly, these complexes were biosynthesised by the Streptomyces by incubation in a ferrous environment (e.g., with FeSO_4_) with carbon sources such as fructose and alanine in an assumingly complex pathway that is not extensively discussed [12], phosphate sources were also provided in most cases [11]. They were eventually identified as secondary metabolites and did not appear to assist in the regulatory uptake of iron into the cell [12]. The same iron complexes have now also been synthesised, using Cronheim’s copper-mediated nitrosation method (to be discussed in due course), followed by transmetallation to the ferrous complex [12,13,15].

### 1.2. A Brief History of Metal-Nitrosophenolato Synthesis

The very first appearance of metal-nitrosophenolato complexes was reported by Eugéne Millon [16] in 1849, involving the formation of a mercury(II) complex. As indicated, such complexes are usually highly coloured, which inspired their continued preparation and isolation; indeed, the first fully isolated example appeared as early as 1900, but was originally misidentified as an *ortho*-nitrophenol complex of mercury with four pendant ligands [17]. It has now been comprehensively shown that this isolated compound was actually the corresponding mercuric-nitrosophenolato complex [18,19]. The analytical value of these intensely coloured complexes has over the years been widely exploited via the ‘Millon reagent’ or ‘Millon test’. This test has been used as a qualitative detection method for tyrosine, as well as other phenol functional molecules present in solution [14,17].

The next big development in the preparation of metal-nitrosophenolato compounds was realized by the Austrian chemist, Oskar Baudisch, during the early part of last century [20,21]. The developed synthesis represents a very interesting and novel functionalisation of an aromatic ring (such as benzene), where a hydroxy and nitroso moiety are added with defined *ortho*-regioselectivity. The functional aromatics are formed in the presence of a metal ion, typically copper(I) or (II) and hence post reaction, the compounds remain co-ordinated to the metal, forming a stable dimeric complex (in the case of copper). Advances in the fundamental understanding of this sequence, which has become known as the ‘Baudisch reaction’, including its underlying mechanism, have been explored in several publications, but the experimental results and conclusions have never been reviewed in a comprehensive fashion, and despite multiple recent advances in copper-mediated nitrosation procedures [22,23], the related Baudisch scheme has been neglected. This review aims to amalgamate existing knowledge in order to highlight the latest thinking regarding the mechanism and disclose how this reaction and its products may find fertile ground as a valuable transformation in modern synthetic chemistry. 

### 1.3. Introduction to the Baudisch Reaction

While not being a commonly encountered or formally recognised named reaction, the term ‘Baudisch reaction’ has become synonymous with specific processes that synthesise metal-nitrosophenolato compounds. In the original reported work, Baudisch employed a range of copper(I) and (II) sources and ‘Merck’s Superoxol’ (historic, commercial name for aqueous hydrogen peroxide) with benzene [20,21]. In addition, either hydroxylamine or a nitrous acid source was further supplied to introduce the corresponding nitroso moiety, hence forming complexes as the end product (Scheme 1 and Scheme 2). Baudisch in his reports proposed that the mechanism was much the same whether hydroxylamine or a nitrite source was used, although findings from later publications to be discussed cast significant doubts over this conclusion. 

### 1.4. Copper-Mediated Aromatic Nitrosation

An alternative, but related scheme that produces metal-nitrosophenolato complexes converging from phenolic starting materials was reported by Cronheim, several years after Baudisch’s original discoveries [15]. In this reaction aqueous nitrosation conditions (nitrite salt with acid) were used, and hence the process most likely occurs via an aromatic *C*-nitrosation, where the copper plays a crucial, yet undisclosed role. This approach has been used to synthesise several derivatives disclosed in other publications and is also therein loosely referred to as the Baudisch reaction [18,24,25]. It should be noted that Cronheim and others’ opinion was that this route was distinct from the Baudisch conditions, as often a different major product is generated under the Baudisch conditions. A representative example which clearly distinguishes between these processes is transposed regioselectivity, as seen in the below transformations (Scheme 2).

### 1.5. Additional Syntheses of Metal-Nitrosophenolato Complexes

A further alternative synthetic approach would be to generate the free 2-nitrosophenol compounds first and then react them in a secondary process with a metal salt (e.g., copper(II) chloride). A copper ion, for example, will readily combine with 2-nitrosophenol under a range of monophasic or biphasic solvent systems, due to the substantial binding affinity [1,15,27]. Indeed, the thermodynamic stability of the copper-nitrosophenolato complexes is such that the biphasic reaction of a ligand in an organic solvent with a copper(II) salt in water is often completed quantitatively in only a few minutes [15]. Free 2-nitrosophenols can be made by partial oxidation of 2-aminophenols, in some cases [28], as well as a simple *C*-nitrosation of a phenol [27,29,30,31]. The scope of this particular approach however appears limited to certain 5-substituted-2-nitrosophenols and *ortho* nitrosonaphthols [1,31,32].

In contrast, the ligands can be freed from the complexes through the action of a strong acid. These reactions were demonstrated by Cronheim to show the formation and decomplexation of these complexes, as visualised by substantial colourimetric differences. The recovered yields for the free ligands were not reported; but Cronheim described ‘relatively low yields’, which is expected due to the nitrosophenol compounds’ strong ligand binding affinity and the temperamental acid-stability of the free ligands. A much improved technique allowing simple isolation of the 2-nitrosophenol involves the use of a copper-scavenger, which coordinates with a higher binding affinity and thus selectively displaces the chelating nitrosophenol ligands [24].

The ligand-metal association method is not as applicable as might first be expected due to problems with isolating the free 2-nitrosophenol compounds. Indeed, there are only very limited experimental procedures documenting the preparation and isolation of 2-nitrosophenols. It is a common misconception that *C*-nitrosation of phenols predominantly produce the corresponding 2-nitrosophenol regioisomer, as per the general expectation for an S_E_Ar process [29,33,34]. Conversely, unless there is a blocking group at the 4-position or a very strongly directing or bulky group at the 3-position, nitrosation of phenols occurs almost exclusively *para* to the hydroxyl group [35]. This occurs because even though the 2-nitrosophenol is the kinetic product, owing to the ability of the nitroso group to undergo migration to a more favourable position of the Wheland intermediate, the thermodynamic 4-substitued product is almost the sole isolated adduct [2,36]. It should also be noted that examples of 4-substituted-2-nitrosophenols are still relatively rare and mainly historical, the one recent publication [37] describing the preparation, isolation and characterisation of four such compounds has since been retracted, hence the existence of these compounds remains unconfirmed. Electronics also play an important role as aromatics with electron-withdrawing substituents are generally inert to nitrosation (hence why poly nitrosation is rarely observed), to a greater extent even than nitration [38], with toluene being about the least electron-rich aromatic known to undergo *C*-nitrosation [28,32,34]. 

This preference for *para*-nitrosation is seemingly altered in the Baudisch conditions where the formation of a complex stabilises the 2-nitrosophenol ligand against rearrangement. Complexation also helps prevent condensation/decomposition or oxidation to the nitro derivative [18,36], factors which likely explain why free 2-nitrosophenol compounds are so rarely reported and characterised in the literature. Despite their highly reactive nature, a small collection of publications describe the synthesis and isolation of 2-nitrosophenols possessing a meta substituent, rather than a *para* ‘blocking’ group (Scheme 3) [31,39,40]. In each case, the metal substituent was a π-donor comparable in strength to or exceeding the phenol directing group, hence attenuating the natural preference for *para*-nitrosation relative to the phenol.

Consequently, not all metal-nitrosophenolato complexes can be formed when metal salts are combined with free 2-nitrosophenols [41]. Several failed attempts to synthesise complexes have been reported [1]; however, investigating the experimental approach it is not clear if this was as a result of complex instability or actually unsuccessful synthesis of the ligand in the first place. In general, there are four principal synthesis routes to the discussed complexes, and although their preparations can be problematic, the corresponding complexes can often be readily obtained by simple precipitation. Some representative reaction conditions are summarised in Table 1 and will be discussed further in more detail subsequently.

### 1.6. Scope of Metal-Nitrosophenolato Synthesis

While the mechanism of the Baudisch reaction implies that almost any aromatic should be able to form a copper-nitrosophenolato complex, in practice, the Baudisch conditions have not been extensively investigated for their scope. More commonly, this approach has been relegated in favour of either the copper (or other metal)-mediated phenol nitrosation procedure or the direct complexation of the free-ligand with a metal salt (it is worth noting that these latter schemes are still often unhelpfully referred to as the ‘Baudisch reaction’). For this reason, there is often a strong correlation between the number of available literature references for a given nitrosophenol ligand and that of the related metal complex. The exception is for 4-substituted-2-nitrosophenols, such as 4-chloro-2-nitrosophenol, where there has been no reliable confirmation of existence for the free ligand, whereas the corresponding metal complex has been repeatedly reported. The sub-class of *o*-nitrosonaphthols appears to form the most popular complexes, probably because several *o*-nitrosonaphthols are more stable and are also commercially available [44]. A range of metals have thus been utilised in their complexation. Historically, these were initially smaller, first-row transition metals, though larger metals have subsequently been found to form complexes too, including (but not limited to) gold [15], palladium [45], ruthenium [46] and more recently even samarium [47]. 

### 1.7. Properties of Metal-2-Nitrosophenolato Complexes

One of the first publications highlighting the properties of a vast assortment of metal-nitrosophenolato complexes came from German chemist, Georg Cronheim [15], whose team prepared and analysed over 50 different complexes. It was also suggested in this seminal publication that the free 2-nitrosophenol ligands were isolated from these complexes; however, yields, purities, stabilities or structural data were not provided to support this statement. It was found that stable complexes formed with a variety of metal +2 centres, especially mercury(II), nickel(II), iron(II), cobalt(II) and palladium(II). Other metals in different oxidation states, including silver(I) and gold(III) were also found to have affinity for the free 2-nitrosophenols. Interestingly, if insufficient free nitrosophenol was present for complete conversion to the *bis*-ligated complexes, mono-ligated complexes were selectively formed. These could be easily distinguished by their different solubilities, because the bi-ligated complexes were organic soluble but not aqueous, and the reverse was true for the mono-ligated. The majority of the complexes reported were prepared using Baudisch conditions; however, as 4-substitued-2-nitrosophenolato complexes cannot be made in this way, they were instead generated using what appears to be the first record of a copper-mediated nitrosation of phenols procedure (excluding examples of the Millon test [16]).

A later study, by Charalambous et al. [1], investigated the properties of additional metal-2-nitrosophenolato complexes and addressed some of the common misconceptions made in the earlier literature. Using infrared spectroscopy, it was discovered that the nitrosophenol ligands readily tautomerise to the more favoured *o*-benzoquinone monoxime structure (Figure 3), and the same is true for the free nitrosophenols [29].

It was not possible at the time to determine whether the metal atom coordinated to the nitrogen or the oxygen atom of the nitroso group. It has now been confirmed using single crystal X-ray diffraction (XRD) analysis that coordination is through the nitrogen (Figure 4) [29,48]. A Jahn-Teller distorted octahedral geometry is adopted, with the two aromatic ligands forming an equatorial plane and the axial positions occupied due to oligomerisation, with the phenolic oxygen typically as the donor, although its distance is a little outside the primary coordination sphere of the copper and instead within the Van der Waal’s radii [49]. This structure occurs for only completely dried, anhydrous compounds, otherwise a water (or another solvate, e.g., ethanol) is found sitting at the apex of the structure in the place of a neighbouring phenol oxygen. Almost all other published structures (where two nitrosophenolato ligands are present, with a single additional small donor) share the back-to-back square-based pyramid geometry, apart from one example, which reports a distorted trigonal bipyramid [50].

In all cases studied, when the copper complex is prepared the system tends to adopt a *bi*-ligated dimer unit as a maximum whereas other metals, such as the higher oxidation states of cobalt(III) or the larger iron (II or III) ions have been known to accommodate an additional nitrosophenol ligand, i.e., retaining an octahedral hexacoordinate structure bearing 3 ligands (Figure 5) [52] (although with nitrosophenols bearing larger substituents, e.g., 4-bromo-2-nitrosophenol, complexes remain *bis*-ligated, at least with Co(III)) [53]. These findings were fully supported by elemental analysis and UV-VIS absorbance data and proved at the time most revealing as up to this stage all complexes were assumed to be limited to just one or two nitrosophenol ligands [3,43].

### 1.8. Derivatisation of Complexes

While the organic nitrosophenolato ligands have enormous affinity for small metallic ions, like Cu(II), researchers have been able to derivatise the complexes with additional, neutral ligands. Cronheim first used pyridine as an auxiliary ligand to alter complex solubility [15], noting that the new compounds could be formed by simply mixing the copper (or other metal) nitrosophenolato complex with pyridine. Castellani et al. also synthesised several new complexes with additional neutral amine bases, again by simply stirring the parent complex at room temperature in acetonitrile, with a slight excess of the Lewis base (**L**, **6a**–**f**, Table 2) [55]. Later, it was discovered that 6-coordinate complexes could be generated by additionally heating the complex with three equivalents of imidazole or *N*-methylimidazole in refluxing acetonitrile (**6g**,**h**, Table 2) [50] in order to overcome the interaction of the copper centre with a neighbouring phenolic oxygen atom. Other notable derivatisations recorded include the synthesis of a potassium (µ-iodo) copper nitrosophenolato complex, which comprises copper in both its octahedral and square-based pyramid geometries, bound by two bridging iodides (**6i**, Table 2) [56].

The complexes have also been shown to undergo transmetalations. Tamura et al. [18] report the conversion of mercury *bis*(4-methyl-2-nitrosophenolato) to the corresponding copper *bis*(4-methyl- 2-nitrosophenolato) in the presence of a simple copper salt (CuSO_4_). Remarkably, Charalambous, Castellani and co-workers have been able to synthesise and characterise crystalline nitrosophenolato complexes prepared from alkaline metals, i.e., sodium and lithium, by reacting with sodium and lithium hydroxide respectively [57]. Due to the redox characteristics of the ligands, these may have interesting properties for areas such as energy storage and battery materials [58].

## 2. History and Development of the Complex Formation

When Baudisch first discovered his named reaction, he proposed a mechanism involving the addition of a ‘nitrosyl radical (NOH)’ and a hydrogen across a double bond of the aromatic ring, promoted by a Cu(I) ion, which was oxidised in the process to the Cu(II) product (Scheme 4). The highly reactive addition product (**1a**, Scheme 4) was then proposed to aerobically oxidise, furnishing the 2-nitrosophenol (**4**, presumably furnishing water as a side-product). The order of reagent addition in Baudisch’s reported reactions (Scheme 1) implies he believed the mechanism proceeded through nitrosation, followed by secondary phenolation. Baudisch stated that the ‘nitrosyl radical’ could be generated in 3 different ways: (1) via oxidation of NH_2_OH with Cu(II), (2) reduction of HNO_2_ with Cu(I), and (3) release from benzenesulfohydroxamic acid by copper ions and H_2_O_2_ [20]. While no radicals of a type corresponding to NOH are known to exist, the nitric oxide (·N=O) radical can be observed under nitrosation conditions, but is rapidly oxidised in the presence of oxygen to the active nitrosation species (nitrosonium, NO^+^) and can itself only form *C*-nitroso compounds by reacting with a carbon centred radical under an inert atmosphere [29]. However, the nitric oxide radical is also known to react with copper ions to form a copper-nitroso complex, itself a powerful agent for nitrosation [29]. Although such a species could be present under the Baudisch reaction conditions and may explain some of the behaviour of the ‘nitrosyl radicals’, there remain several issues with Baudisch’s proposed sequence. Firstly, the hydrogenated intermediate (**1a**, Scheme 4) is not accounted for: if benzene attacks the nitroso ligand of a copper-nitroso complex, then the next stage would be deprotonation to restore aromaticity, rather than hydrogenation to disrupt it further by generating an undesired, higher-energy intermediate. It also seems doubtful that the nitrosation would occur before hydroxylation, where the introduction of the hydroxyl group would result in activation of the ring towards subsequent nitrosation. It is well documented that direct nitrosation does not occur on unsubstituted benzene [33]. Despite the issues with the mechanistic aspects of the reaction, based on little more than theory and colourimetry, Baudisch correctly hypothesised the final copper-chelated product (**2**).

### 2.1. Development of the Baudisch Reaction Mechanism

In 1955 Konecny [43] reinvestigated the Baudisch reaction conditions and expanded upon the work using different metal salts of Cu(I), Fe(II), and elemental Cu(0); all of which were found to form the desired complexes. Interestingly, traces of 2- and 4-nitrosophenol (major) were detected in reactions involving the mixing of benzene, H_2_O_2_, HCl, and H_2_NOH (without a metal). Irradiation of the reaction mixture with X-rays was also found to increase the rate of the metal-free reaction. In a control reaction, performed by replacing benzene with phenol, the metal-free reaction took place readily (generating 4-nitrosophenol as the major product), suggesting a rate-limiting hydroxylation step.

Konecny noted that although there was no direct evidence for the formation of ‘NOH’ radicals, there was evidence for the formation of hydroxyl radicals when employing H_2_O_2_ with Fe(II) salts, copper metal, or with exposure to short-wave (X-ray) radiation [43]. These conditions were known to convert benzene to phenol, and as phenol had been shown to react with hydroxylamine and H_2_O_2_ (without a metal) to yield 4-nitrosophenol, this gave a more rational sequence of events. The reaction was therefore considered to progress through an initial hydroxylation followed by nitrosation. The nitrosating agent was proposed to be nitrous acid (while nitrous acid is a precursor to, but not a nitrosating agent, at the time nitrous acid was thought to be an active nitrosating agent) formed as an intermediate in the oxidation of hydroxylamine by H_2_O_2_ (or alternatively from K/NaNO_2_ and acid). Supporting evidence also came from experiments involving related copper-mediated nitrosation of phenol (utilising NaNO_2_, H_2_SO_4_ and Cu(II)), which yielded a ‘red’ complex [43]. 

The refined mechanism suggested by Konecny became the accepted sequence, until a systematic investigation by Maruyama et al. [26,43] identified several inconsistencies. The first major concern was the nature of the nitrosating agent, as a free nitrosation agent does not explain the observed preference for -*ortho* regioselectivity. A new proposal was therefore put forward that a copper complex of hydroxylamine was involved in the nitrosation. As was shown, mixing anhydrous Cu(II) salts with hydroxylamine hydrochloride in methanol produced a complex with a 2:1 hydroxylamine:Cu(II) stoichiometry [42]. Mixing this complex with phenol and H_2_O_2_ gave exclusively 2-nitrosophenol (derived from decomplexation), implying that the hydroxylamine complex could be a precursor to an intermediate complex involved in the Baudisch reaction.

Another observation was that at higher pH (>4) than the standard Baudisch reaction (pH 2–3), the yield of the 2-nitrosophenol (**4**) was considerably lowered, and instead, mainly catechol was formed. Given normal hydroxylation of phenol would be expected to preferentially produce 1,4-dihydroxybenzene [28], the suggestion was made by Maruyama et al. that the two products were produced through the same intermediate complex (**7a**, Scheme 5), and that the outcome was thus pH dependent.

The re-examination by Maruyama et al. of the reactions of certain phenols produced some additional surprising results [42]. For example, the reaction of 1-naphthol (**8**) gave, as predicted, the 2-nitroso-1-naphthol adduct **9**, but the reaction of 2-naphthol (**8a**) also gave the same 2-nitroso-1-naphthol product (**8c**, Scheme 6). This apparent displacement of a hydroxyl group inspired a wider examination of other phenolic starting materials. The compound 4-methylphenol (*p*-cresol, **3a**, Scheme 6) was converted to 5-methyl-2-nitrosophenol (**4a**) and gave the same product as starting from 3-methylphenol (*m*-cresol, **3b**). Again, this implied exchange of the hydroxyl group had taken place. Consequently, catechol was re-examined, and it was found under the original Baudisch conditions to yield the 2-nitrosophenol.

Clearly, the reaction was therefore not quite as simple as the proposed reaction pathway outlined in Scheme 4 suggested. The existence of a catechol intermediate demonstrates that the mechanism of the Baudisch reaction with hydroxylamine must differ from the simpler copper-mediated nitrosation of phenol, which does not accept catechol as a substrate, nor involve replacement of any existing phenolic hydroxyl groups [16]. For a hydroxyl substitution to take place, the reactions must go through a common intermediate with two *ortho* oxygen atoms presumably chelated to the copper, similar to the intermediate suggested from the reaction at pH > 4 (**7iii**, Scheme 5 and Scheme 6).

Additional experiments performed by Maruyama et al. convincingly identified the existence of a catechol-type intermediate, but the question remains as to how the aromatic hydroxyl group can be replaced when neither nucleophilic nor electrophilic substitutions at a phenolic carbon are favourable processes. An alternative possibility would be via an oxidised *o*-quinone intermediate, to which nucleophilic addition is possible, especially if the copper acts as a Lewis acid, withdrawing electron density and thus activating the quinone carbonyl π-bond. To investigate this possibility, the reaction was attempted with semicarbazide hydrochloride in place of the hydroxylamine [42]. A mixture of catechol, Cu(II) and semicarbazide hydrochloride (at pH 2.5) did not react until H_2_O_2_ was added, confirming the reaction was of the same type as the Baudisch reaction. The resulting product (determined by elemental analysis) was identified as 2-hydroxyphenylazoformamide, the semicarbazone of *o*-benzoquinone (Scheme 7).

In addition, when 4-methylpyrocatechol (**9a**) was reacted in the same manner, it gave exclusively 4-methyl-2-hydroxyphenylazoformamide (Scheme 7). The compound was confirmed by decomplexation and reduction by SnCl_2_ to the corresponding aniline, 6-amino-3-methylphenol, for comparison (Scheme 7). The observed regiochemistry was identical to the outcome of the original Baudisch reaction. Since semicarbazones are formed by reaction of carbonyls with semicarbazides, the conclusion was that this indicated the existence of an *o*-quinone intermediate [42]. This leads to the following sequence of events: firstly, the intermediate catechol-Cu(II) complex (**6c**) is oxidised to an *o*-benzoquinone derivative which then reacts to yield the semicarbazone (with semicarbazide) or the oxime (with hydroxylamine). In line with this conclusion, 2-benzoquinones gave 2-nitrosophenols under the general conditions of the Baudisch reaction without the need for the addition of H_2_O_2_. As a final confirmation of the sequence of steps, all catechols tested did not react until H_2_O_2_ was added as a co-oxidant [42].

### 2.2. The Accepted Mechanism

The new Maruyama proposition for the Baudisch conditions as applied to benzene while acknowledging all the preceding experimental findings is outlined below. Starting from non-phenolic starting materials, the process begins with a copper(I)-catalysed fragmentation of hydrogen peroxide, generating hydroxyl radicals that rapidly attack the aromatic ring to form phenols (Scheme 8).

Next, coordination of the newly formed intermediate phenol to the copper occurs, which is also complexed with two hydroxylamine ligands and a peroxyl ligand (Scheme 9). Attack from the peroxyl ligand follows, forming the coordinated catechol intermediate. An oxidation sequence then forms the 2-benzoquinone complex. This key intermediate enables nucleophilic attack by a hydroxylamine ligand, leading to substitution. This substitution occurs unusually at the seemingly most electron-rich of the two quinone-carbonyls, forming a single regioisomer [42]. Kinetically, attack at the less electron-rich carbonyl would be preferred, as it is more electrophilic, suggesting thermodynamic control. Further evidence for a benzoquinone intermediate is provided in a more recent piece of work that describes the reaction of a 1,2-benzoquinone with hydroxylamine in the presence of copper, forming the target copper complex [59].

Hydroxylamine is readily oxidised to higher *N*-oxidation levels in the presence of strong oxidants including H_2_O_2_ [60], so a legitimate question is how the hydroxylamine survives long enough for this activity. It is possible that the complexation of the hydroxylamine protects the hydroxylamine (the hydrogen peroxide is the final reagent added) and the *N*-oxidation level only increases once bound to the ring, in order to restore aromaticity [42]. However, the above mechanism does not account for the fact that the process still works when KNO_2_ is used in place of hydroxylamine (as reported in the original Baudisch work). Under such alternative conditions, the higher nitrogen oxidation level excludes a nucleophilic attack. It is known that when sodium nitrite is mixed with copper(II), a copper-nitroso complex forms, which is a powerful electrophilic, but not nucleophilic, nitrosation agent [29]. Although it may be possible that such a copper-nitroso complex undergoes a redox process, facilitating the oxidation of phenol/catechol to the intermediate *o*-benzoquinone with concurrent reduction of the nitroso ligand to a more nucleophilic species, this seems unlikely. Alternatively, it is more probable that these reactions performed by Baudisch using KNO_2_ rather than hydroxylamine do not ultimately follow the same outlined (Maruyama’s) mechanism. Indeed, they are likely to be more closely related to the copper-mediated nitrosation (see above Section 1.4), and the role of the hydrogen peroxide was for the formation of phenol from benzene. This then represents an alternative—but related—one-pot sequential process starting from benzene. 

### 2.3. Mechanism of the Alternative Copper-Mediated Nitrosation

Both Cronheim and Maruyama have shown that some phenols (e.g., 4-methylphenol) give a different major regioisomer depending on whether Baudisch conditions with hydroxylamine or copper-mediated nitrosation conditions are used [15,42]. What is less certain is how copper-mediated nitrosation of phenols compares to the equivalent process in the absence of copper. Modern knowledge assumes a rate-acceleration in the presence of copper, again due to the formation of a stronger nitrosation agent than free nitrosonium [29], but it is not immediately obvious from the literature whether the regioselectivity and preference for 2-nitrosophenols is affected.

Results from Maruyama suggest that the presence of copper only has a very minor effect on the regiochemistry versus nitrosation in the absence of copper and that *ortho*-nitrosation dominates only if the Baudisch conditions with hydroxylamine are employed (Figure 6) [42]. However, Konecny repeated the same reaction and claimed that the 2-nitrosophenol adduct (as the copper complex) dominated, [43] albeit without quantification; this appears to be a direct contradiction of the literature. 

The most stable free nitrosophenol regioisomer is the 4-nitrosophenol, because the 4-substituted Wheland intermediate is thermodynamically favoured compared to the 2-substituted (while the 3- substituted is very high-energy due to inability to form quinone-type resonance structures or the ability of the OH to stabilise the initially formed positive charge) [60]. In the Wheland intermediate with *para* regiochemistry (4-substituted), the phenol group can stabilise the electron-deficient nitroso-bound carbon by donating π-electron density more effectively than if the nitroso were *ortho*-bound [38]. This is more significant than the stabilisation gained from the formation of an intramolecular hydrogen bond when the hydroxyl and nitroso groups are adjacent [61]. The effect of other substituents, such as an alkyl, is generally inferior to that of the phenol group, although it is undoubtedly observed when not in competition with more powerful effects. For example, one of two regioisomers forms when reacting 4-methlyphenol under Baudisch conditions; hence, the nitroso is found exclusively *para* to the methyl, despite the need for the original phenol oxygen to be replaced by the nitroso. Additionally, only one of two possible regioisomeric products from the reaction of 3-methylphenol is formed (Scheme 10) [42], again the resulting nitroso is *para* to the methyl. As previously discussed, the mechanism of nitrosation allows the most stable regioisomer to dominate [33,36]; hence, an outcome where the resultant nitroso group is not *para* to the phenol would qualify as an unexpected result. In addition to Konecny’s paper, there exist several other publications [1,25] that quote a major regioisomer that might be unexpected if the copper did not influence the regioselectivity in some way (Table 3).

Literature results can be found to both support and rule out the possibility of copper playing a key role in forcing nitrosation of phenols to occur predominantly at the *ortho* position rather than *para* position (Table 3); thus, reviewed together, the verdict remains inconclusive. To help resolve this, our group has performed several copper-mediated nitrosation reactions [41], using similar conditions to those first devised by Cronheim [15]. For all reacting phenolic substrates, including phenol (**7**), which are known to form 4-nitrosophenol in the absence of copper, a highly coloured copper compound was isolated in high yield (Figure 7). The isolated compounds were all initially believed to be copper *bis*(nitrosophenolato) complexes, causing the belief that the copper does cause *ortho*-nitrosation to dominate on substrates where *para*-nitrosation would otherwise be expected. However, detailed analysis showed that the reaction of phenol (**7**) instead generated a stable complex of the *para*-nitrosation product. While it is easy to envisage how the literature disagreement occurred, our results confirm the proposition by Maruyama et al. that copper-mediated nitrosation is not inherently *ortho*-regioselective, as per the Baudisch reaction. 

Since the mechanistic investigation by Maruyama and co-workers, there have been only minimal attempts to further investigate or improve on the mechanism of either the Baudisch reaction or the simpler copper-mediated nitrosation procedures. Instead, researchers have found a variety of unique and innovative uses for the metal complexes. Professors emeriti Carla Bisi Castellani of the University of Pavia [64] and John Charalambous of London Metropolitan University [65], in particular, have contributed substantially to this field, with several publications scattered across the final quarter of the 20th century.

## 3. Applications of Copper-Nitrosophenolato Complexes

An obvious application of these complexes is as the means to synthesise problematic 2-nitrosophenol compounds. The ease of oxidation of the often-unstable free nitroso compound to a more stable nitro derivative offers a mild synthetic route to such compounds. Nitrophenols have a wide range of applications with the nitro groups being useful scaffolds, via reduction to the amine followed by dediazoniation [66] and modification (e.g., Sandmeyer processes) [67]. Although the production of 2-nitrosophenols via these sequences are stoichiometric in copper, a highly efficient process could be envisaged if the metal ion can be recycled. In the literature, the copper-mediated nitrosation has been used preferentially relative to the Baudisch conditions, presumably because of its simplicity and often higher isolated yields.

### 3.1. Colourimetry

As we highlighted previously, the very first reported preparation of a metal-nitrosophenolato complex was in the detection of tyrosine, by utilising the intense colour of the mercury complex in solution [16]. Subsequently, 2-nitrosophenol products have been more widely used in colourimetry to identify metal ions, as originally suggested by Baudisch; for example, in the determination of Co(II) ions [15]. The Co(II) complex of 2-nitrosophenol is a grey/brown colour and is only sparingly soluble in petroleum ether. The high affinity of 2-nitrosophenols for Co(II) is such that essentially 100% of the metal ion is extracted from water upon shaking with a solution of 2-nitrosophenol in petroleum ether, giving a grey/brown solution, which can be easily measured in a spectrophotometer. The colour is stable for ‘at least several hours’, and can be detected down to a concentration of 0.1 ppm [15].

Dessouky [25] expanded the general principle of the Millon test by using the copper-mediated nitrosation on target samples containing phenylephrine (**3e**, Figure 8), and subsequently measured the absorbance of the resulting copper complex in solution. It was found that even in pharmaceutical preparations containing additional formulation materials, the correct concentrations could be determined to a high accuracy. When reacted, the phenylephrine gave a coloured complex, which lasted for up to 24 h with a minimal decrease in absorbance. Other phenolic materials of pharmaceutical interest were also tested, including adrenaline, tetracycline and methyl salicylate, although none gave the same long-lasting red colour [25]. In addition, it was claimed that the formation of the complexes was close to quantitative, although speculation about the reactivity of the secondary amine with NaNO_2_, forming a nitrosamine, as well as the possible tendency for the phenolic hydroxyl to give *para* directed nitrosation, casts some doubts on the likelihood that this formation could be completely lossless. Such analytical tests have now been somewhat superseded; therefore, to the modern synthetic chemist, the following synthetic procedures will be of more interest.

### 3.2. Nucleophilic Addition of Grignard Reagents

Mustafa and Kamel [68], investigating reactions of Grignard reagents and exploring the chemistry of *o*-quinone-monoximes, found that unusual secondary hydroxylamines could be generated. The nucleophilic addition took place on the N=O bond of the nitrosophenol tautomer, rather than the C=O, owing to the greater electrophilicity of the *N*-atom. Phenanthroquinone monoxime (**10**, Figure 9) was used as a substrate, and several Grignard reagents were tested, including phenylmagnesium bromide, for example, which gave 10-phenylhydroxylamino-9-hydroxyphenanthrene (**10a, R=Ph**) after work-up (Figure 9). In addition, reduction of the same starting material, phenanthroquinone monoxime (**10**) was attempted using thiophenol, but instead of the expected reduction product (**10b**), the diphenanthro-oxazine (**10c**) was synthesised via a rapid exothermic reaction (Figure 9). The reaction, conducted in cold benzene, this time gave a mixture of the diphenanthro-oxazine (**10c**) product and the expected reduction product (**10b**), suggesting that **10c** arises from the self-condensation of product **10b** with starting material **10a** [68]. By contrast, the reduction performed with LiAlH_4_ gave solely the 10-amino-9-phenanthrol (**10b**), implying the condensation is not effective under these conditions. The product from the oxidation of the thiophenol is not reported, but is likely the disulfide (PhS-SPh) [69].

### 3.3. The Use of Complexes as a Redox Catalyst

A further interesting application of 2-nitrosophenols was proposed by Charalambous et al. [30] and later expanded by Nishino et al. [70], then Ogura et al. [71]. Noting that manganese plays a vital role in oxygen evolution in the photosynthetic pathway of many plants (during the synthesis of glucose) and that the redox couple nitroso/nitro had been used for oxidation of alkenes to aldehydes and ketones, the researchers attempted using 2-nitrosonaphthol-manganese complexes under an oxygen atmosphere to perform the epoxidation of alkenes (Figure 10) [30].

The majority of reported and industrially utilised catalytic epoxidation systems require the use of active oxygen sources such as iodosobenzene (Ph-I=O), organic peroxides or hypochlorite salts; however, these all produce stoichiometric amounts of waste, leading to poor atom economy and the need for extra product separation/extraction. Alternatively, the reported nitrosonaphthol-manganese system represents a successful, non-porphyrin, alkene epoxidation catalysis using simple ambient oxygen as the oxidant. Both Mn(II) and Mn(III) complexes (with two and three 2-nitrosonaphthol ligands, respectively) were used, and two *ortho*-nitrosonaphthol regioisomers (**11, 11a**) were tested (Figure 10). The general procedure was very mild, requiring only the stirring of the catalyst and alkene in toluene, under an oxygen atmosphere at 60 °C (Figure 10) [30].

Using the same manganese-nitrosonaphtholato complexes, the scope of this process was broadened to include some novel phenolic oxidations as a method of aryl-aryl coupling, giving diaromatic products (e.g., **12a**–**12d**, Table 4) [70]. The presence of trialkylphosphine was required as an additional π-donor ligand, hence allowing a complex to form with a peroxide ligand upon reaction with molecular oxygen. The regioselectivity is ensured by using substrates with R-groups to the *ortho* positions of the phenol. 1,4-benzoquinones were formed as side-products in some cases [70].

Metal-nitrosophenolato complexes have found further use as homogeneous co-catalysts in electrochemical carbon-dioxide reduction, producing methanol [71,72]. The application arises because, as with other metal-*bis*(nitrosophenolato) complexes, there is the vacant coordination site that weaker donors, such as water and organic solvents, can occupy with high reversibility. In this case, the labile CO_2_ ligand binds; hence, the complex provides a route by which CO_2_ can intercede a heterogeneous reduction catalyst. Ogura and co-workers demonstrated this as a proof of concept using cobalt *bis*(2-hydroxyl-1-nitrosonaphthalene-3,6-disulphonato) and a platinum heterogeneous catalyst [71]. Later, the work was expanded by building a functioning, continuous hydrogen fuel-cell that converted CO_2_ to methanol catalytically and indefinitely, provided the pH was controlled and sufficient H_2_ was present (Figure 11) [72]. The implementation of the process to remove unwanted CO_2_ and CO during ammonia synthesis in the Haber process was also established [73], and finally, a solid-supported electrode of the same cobalt complex and platinum was built to improve performance [74]. 

### 3.4. Use in [4 + 2] Cycloadditions

The copper nitrosophenol complex constitutes a heteroatomic diene system and hence can undergo Diels-Alder cycloadditions with powerful dienophiles, such as dimethylacetylenedicarboxylate (DMAD, **13**) [75], to yield unique diheterocycles (**14a**–**14g**, Table 5). A concise report was published by McKillop and Sayer [76], quoting yields as high as 98% (Table 5, R = 4-Me). The process is promoted due to the benzoquinone tautomer, rather than the nitrosophenol and was supported by the lack of product formation when complexes of compounds known to exist predominantly in the nitrosophenol form were used. Interestingly, no reaction occurred between **13** and isolated (non-complexed) *o*-quinone-monoximes (2-nitrosophenols). In addition to stabilising the organic ligand reagents, it was suggested that the Cu(II) plays a twofold role in the mechanism. Firstly, the electron density is polarised towards the two heteroatom termini of the diene system, and secondly, it provides a ‘coordinative template’ for the reaction (the acetylene is believed to weakly coordinating). This process, over two steps, leads directly to a unique bicyclic system, for which derivatives have been investigated for biological properties including as herbicides [76]. Here we note that we repeated this procedure ourselves and the structure of the heterocycle was slightly different to that in the publications [41].

### 3.5. Synthesis of Phenazines

Pathways to more complex structures starting from copper-nitrosophenolato complexes have been shown in a study by Charalambous et al. [35,77,78], who were aiming to prepare nitrene metal complexes via deoxygenation of 2-nitrosophenols with PPh_3_. While nitrenes were proposed intermediates in the reported reactions, they were not the isolated products. Various metal complexes of 2-nitrosophenols were treated with PPh_3_ in chloroform or pyridine, and the products varied with the metal ion in question (Figure 12). The 2-nitrosophenol complexes of Ni(II) or Zn(II) gave triphenyl(2-hydroxyphenylimino)phosphorane metal complexes, whereas Cu(II) and Fe(III) complexes gave 1,6-dihydrozyphenazines (**15**), which are highly desirable systems.

The different outcomes result from the fact that Cu(II) and Fe(III) ions are easier to reduce. The reduction of such metal(2-nitrosophenolato)*_n_* complexes readily generates a nitrene at the nitroso-*N* of one ligand, leaving behind metal(2-nitrosophenolato)*_n-1_*(PPh_3_)_2_. The nitrenes then dimerise to yield 6,10-dihydroxyphenazines. With the Ni(II) and Zn(II) species, reduction of the metal ion is much more difficult, and the PPh_3_ instead substitutes the oxygen of the nitroso group, yielding complexes of triphenyl(*o*-hydroxyphenylimino)phosphoranes [35].

Free 2-nitrosophenols also react with PPh_3_ to furnish 1,6-dihydroxy-5,10-dihydrophenazines (**15**), with traces of the PPh_3_ substitution product. The yields varied (Figure 12), but products were easily separated from starting material. This reaction is different to those which lead to a *O*,*N*-diheterocycle in the cycloaddition processes, as only the *N*-containing functional groups are involved and the hydroxyl groups are outside the newly formed ring [77]. The reported yields were similar starting from either the free nitrosophenol or copper-nitrosophenolato complex.

### 3.6. Synthesis of Oxazoles

Castellani and co-workers combined theory and practice from the previous examples of phenazine synthesis from Charalambous and cycloadditions from McKillop in order to synthesise valued benzoxazoles (**16**, Scheme 11) directly from the copper-nitrosophenolato complexes [75]. The conditions and reagents required are much the same as per those for the aforementioned cycloaddition, with one important difference: the reaction is performed in anhydrous solvent (methanol). The possible mechanism was not elaborated and does not appear to have been revisited since; however, it was thought to be comparable to the reduction-nitrene-formation observed in the phenazine synthesis [75,77]. The reaction time is fairly short (1 h), but forcing conditions were required (Scheme 11).

## 4. Future Applications of Copper-Nitrosophenolato Complexes

### 4.1. Functionalisation of Natural Products

Phenolic moieties are widespread across many natural products; notably in polyphenols, a class of phenol-containing macromolecules prevalent in plant material (e.g., making up about 30% of the dry mass of tea) [79]. With the selective addition of a nitroso group (or nitro via oxidation) to the phenolic moieties, a valuable degree of functionality can be added. The investigation by Dessouky showed that the process works on some such compounds [25], but this could be taken much further. Applications from 2-nitrosated natural products may arise due to their high affinity for metal ions, giving applications in areas for example like removing transition metals from laundry in the washing process (Fe in blood and Cu in tea). 

### 4.2. Lignin Functionalisation

Lignans, the phenolic monomers of lignin, are a promising replacement source of aromatic compounds, currently obtained from crude oil [80,81,82]. There are several reported methods for breaking the lignin down into useful monomeric aromatics (lignans) [83,84], including a recently developed method of hydrogenolysis of the lignin polymers under mild conditions, with the feedstock of ‘kraft lignin’ (a feedstock readily obtained as waste from the paper industry) [85,86]. Being phenolic, the lignans are electron-rich, aromatic systems; good for electrophilic substitution chemistry, but not much more. By preforming a Baudisch reaction (or copper-mediated nitrosation) on the lignans, a nitroso group could be introduced, neighbouring a hydroxyl, under relatively mild conditions. This opens the possibility of the reactivity discussed above, but perhaps more industrially important are the oxidation products, the 2-nitrophenols. The previously electron-rich system is transformed into an electron-deficient system, and a whole new reactivity pattern emerges.

### 4.3. Addition of Nucleophilic Amines to Aromatic Rings

Maruyama et al. demonstrated that Baudisch conditions could be used to replace a catecholic OH group with a nitrogenous nucleophile [42]. The goal was to aid with mechanistic understanding, but this is an outstanding transformation of an inert group. Even today, the nucleophilic substitution at a phenolic carbon remains niche and generally requires a stoichiometric amount of a reagent to sufficiently activate the phenol as a leaving group [87,88,89]. While catechols and other biphenols have been aminated using ammonia-water under high temperatures [90] since mid-last-century, and more recently this has been applied to the addition of an *N*-formyl moiety [91], the Baudisch reaction could potentially be applied to a more diverse range of nucleophilic amines, as well as avoiding the extreme temperatures required for the transformation. Such an approach could be used to synthesise targets such as benzoxazoles [92], as well as an enormous range of active-pharmaceutical ingredients relying on the 1,2-aminophenol synthon, in comparatively few steps, though as of yet there seem to be no recorded attempts at this approach.

### 4.4. Summary of Uses

As discussed, the title compounds have found uses in the determination of metal ion concentrations, epoxidation and redox catalysts, as well as being good reagents for the synthesis of four distinct heterocyclic systems from simple aromatic starting materials. Despite its ‘amazingly simple’ [20] functionalisation of aromatic (particularly phenolic) substrates and well-defined regiochemistry, these processes do not appear to have been investigated further, at the time of writing. This may be attributed to a lack of general understanding in the formation of the copper complexes and their nitrosophenol ligands. In attempting to demystify the procedure and the obscurity surrounding the Baudisch reaction and its related nitrosation process, additional potential uses can be readily theorised.

## 5. Conclusions

The challenges in synthesis, lack of profusion in literature and unique properties of 2-nitrosophenols and their metal salts make them a stimulating target for further study. Across several publications, greater understanding has developed about their properties and the mechanisms to their formation. Applications are demonstrated, but not yet developed—the speciality of the process may afford application in the future. Doubts over specific aspects of the mechanisms proposed remain, and further study and characterisation of the structures proposed would aid in the understanding of the field. A publication [41], completed by our group, represents the initial stages of such as study.

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
