# Peer review of "The Synthesis and Utility of Metal-Nitrosophenolato Compounds—Highlighting the Baudisch Reaction"

_molecules, 2019, doi:10.3390/molecules24224018_

Round 1

Reviewer 1 Report

This is an excellent and instructive review of a sadly neglected field that has much potential and deserves to be revived. The authors have done a great job at explaining the relevance of o-nitrosophenols in modern chemistry and the complications associated with the various conditions that can be used to generate o-nitrosophenols. This reviewer feels that this review will help rekindle the interest in the compounds/reactions and has only a few suggestions to improve the manuscript.

1. In section 1.1, it would be interesting to briefly discuss the biosynthesis of these antibiotics to offer a point of comparison with the chemical methods outlined afterwards.

2. In section 1.3, reactions #2,4,5: the compound CuOH is shown above the arrows. Cuprous hydroxide is remarkably unstable, should't Cu(OH)2, Cu2O or CuO be shown above the arrow?

3. As a general note, room temperature should be abbreviated "rt", "r.t." or "RT" throughout the manuscript instead of the novel "room T"

Author Response

We gratefully acknowledge the positive comments made by reviewer 1, as well as helpful points to improve the clarity of the script, which we have addressed as followed.

Reviewer 1 suggested “In section 1.1, it would be interesting to briefly discuss the biosynthesis of these antibiotics to offer a point of comparison with the chemical methods outlined afterwards”. We have added some additional text further describing the synthesis and disclosed function of the iron complexes. Reviewer 1 pointed out “In section 1.3, reactions #2,4,5: the compound CuOH is shown above the arrows. Cuprous hydroxide is remarkably unstable, shouldn't Cu(OH)2, Cu2O or CuO be shown above the arrow?”. We agree that it was unlikely cuprous hydroxide (CuOH) could have actually been used, but it is nevertheless the substance that the literature source clearly states. We have added an explanation to the footer of this table to avoid confusing the reader ‘Baudisch states in the literature that ‘freshly prepared yellow cuprous hydroxide’ was used; it is known this is not a stable entity, so the actual active substance could well be an alternative such as cuprous oxide, Cu2O’ . Reviewer 1 suggested “As a general note, room temperature should be abbreviated "rt", "r.t." or "RT" throughout the manuscript instead of the novel "room T"”. We have thus replaced all occurrences with a corrected abbreviation, r.t. (four occurrences).

Reviewer 2 Report

The authors (A. J. Nicholls, et al.) report a review of the syntheses and reactions using Baudisch reaction.  The Baudisch reaction is one of the classical synthetic methods for nitrosophenols using metal ions.  It is interesting the paper focused on the recent trends of the reaction through the past.

Totally, I think the manuscript is well organized and suitable for publication on Molecules after minor issues as listed below.

Line 47 on page 2, correct the name of ‘Viridomcycin A’ to ‘Viridomycin A’. In Fig 2, the figure and figure title should be written on the same page. Line 71 on page 3, ‘ortho-’ should be written in the italic face. Line 76 on page 3, delete the space between ‘[14,’ and ‘17]’. In Scheme 2, the scheme should be drawn on the same page. In Scheme 2, the scheme number should be written in the bold face. Line 155 on page 5, delete the space between ‘[28, 32,’ and ‘34]’. In Table 1, the table title should be aligned with the left. In Table 1, the table should be drawn on the same page. Line 217 on page 8, there are two ‘Figure 2’. Delete the one and renumbering all figure numbers after the page in the manuscript. In Table 2, the table title and the footnote should be aligned with the left. In Table 2, the table should be drawn on the same page. In Scheme 5, the compound’s numbers in the scheme title should be written in the bold face. In Scheme 6, is the compound’s number of ‘2-nitroso-1-naphthol’ ‘8c’? In the text, the authors showed as ‘9’ (see line 344 on page 12).  The authors should check all product numbers and be corrected them. In Scheme 6, ‘o-’ and ‘m-’ cresol should be written in the italic face. In Scheme 8, the scheme should be drawn on the same page. In Table 3, the table should be drawn on the same page. In Figure 6, the figure number should be written in the bold face. Line 496 on page 19, are the references necessary? If they are not necessary, delete them from the text and References, and then renumbering the reference numbers. Line 550 on page 20, is the starting material ‘10a’? Not ‘10’? Line 557 on page 20, the subtitle should be written by using capital letters such as ‘The Use of Complexes as a Redox Catalyst’. Line 571 on page 21, ‘Figure 12’? Not ‘Figure 9’? Line 622 on page 23, delete the period in front of the section number, ‘.3.5’. Line 628 on page 23, ‘Figure 14’? Not ‘Figure 11’? Line 630 on page 23, the compound’s number should be written in the bold face. In Figure 11, ‘o-’ hydroxyphenyl… should be written in the italic face. Line 727 on page 26, the characters in the author’s name are corrupted. Line 783 on page 27, show as a subscript in the number of ‘Mo(O)2(acac)2’. Line 790-791 on page 27, delete the spaces in ‘5 - Substituted’, ‘2 - Nitrosophenol’, and ‘2 - Aroylbenzoxazoles’. Line 797 on page 28, delete the spaces in ‘tri fl uoroacetic’. Line 807 on page 28, delete the spaces in ‘tri fl uoroacetic’. Line 835 on page 28, delete the ‘<scp>’ and ‘</scp>’. Line 840 on page 29, show as a subscript in the number of ‘Cu(Clqo)2’. Line 893 and 898 on page 30, show as a subscript in the number of ‘CO2’.

Author Response

We gratefully acknowledge the positive and constructive comments made by reviewer 2 and especially appreciate the attention to detail and quality of the reviewing that has taken place, allowing substantial improvement to the script. We have addressed these findings as follows:

Reviewer 2 noted “Line 47 on page 2, correct the name of ‘Viridomcycin A’ to ‘Viridomycin A’”. This has been corrected. Reviewer 2 pointed out “In Fig 2, the figure and figure title should be written on the same page”. This has been corrected. Reviewer 2 noted “Line 71 on page 3, ‘ortho-’ should be written in the italic face”. This has been corrected. Reviewer 2 suggested “Line 76 on page 3, delete the space between ‘[14,’ and ‘17]’”. This reference is now formatted correctly. Reviewer 2 noted “In Scheme 2, the scheme should be drawn on the same page. In Scheme 2, the scheme number should be written in the bold face”. We have corrected these issues with scheme 2. Reviewer 2 suggested “Line 155 on page 5, delete the space between ‘[28, 32,’ and ‘34]’”. We have resolved the issue with this reference. Reviewer 2 suggested “In Table 1, the table title should be aligned with the left. In Table 1, the table should be drawn on the same page”. We have resolved these issues with table 1. Reviewer 2 pointed out “Line 217 on page 8, there are two ‘Figure 2’. Delete the one and renumbering all figure numbers after the page in the manuscript”. We have renumbered the figures accordingly. Reviewer 2 noted “In Table 2, the table title and the footnote should be aligned with the left. In Table 2, the table should be drawn on the same page”. We have resolved these issues with table 2. Reviewer 2 pointed out “In Scheme 5, the compound’s numbers in the scheme title should be written in the bold face”. This has been corrected. Reviewer 2 noted “In Scheme 6, is the compound’s number of ‘2-nitroso-1-naphthol’ ‘8c’? In the text, the authors showed as ‘9’ (see line 344 on page 12). The authors should check all product numbers and be corrected them”. The compound numbers have been revised accordingly. Reviewer 2 suggested “In Scheme 6, ‘o-’ and ‘m-’ cresol should be written in the italic face”. This has been implemented and we also corrected the incorrect spelling of nitrosonaphthol on one occasion. Reviewer 2 pointed out “In Scheme 8, the scheme should be drawn on the same page”. This has been rectified. Reviewer 2 noted “In Table 3, the table should be drawn on the same page”.This has been resolved. Reviewer 2 pointed out “In Figure 6, the figure number should be written in the bold face”. This has been corrected. Reviewer 2 commented “Line 496 on page 19, are the references necessary? If they are not necessary, delete them from the text and References, and then renumbering the reference numbers”. We believe these references are necessary to show that the displayed mechanism follows sufficient literature precedent and the references will allow a reader to investigate the mechanism of metal-free C-nitrosation as the mechanism is not largely elaborated within this text. Reviewer 2 commented “Line 550 on page 20, is the starting material ‘10a’? Not ‘10’?”. We believe 10 is the starting material and 10a is the product(s) from reacting with Grignard’s reagents and 10 is the starting material of the second stage too. Reviewer 2 noted “Line 557 on page 20, the subtitle should be written by using capital letters such as ‘The Use of Complexes as a Redox Catalyst’”. We have corrected this. Reviewer 2 pointed out “Line 571 on page 21, ‘Figure 12’? Not ‘Figure 9’?”. This error has been corrected. Reviewer 2 noted “Line 622 on page 23, delete the period in front of the section number, ‘.3.5’”. This typo has been corrected. Reviewer 2 pointed out “Line 628 on page 23, ‘Figure 14’? Not ‘Figure 11’?”. This error has been resolved. Reviewer 2 noted “Line 630 on page 23, the compound’s number should be written in the bold face”. This has been corrected. Reviewer 2 pointed out “In Figure 11, ‘o-’ hydroxyphenyl… should be written in the italic face.” This has been resolved. Reviewer 2 noted “Line 727 on page 26, the characters in the author’s name are corrupted”. This is referring to the reference of the AATCC. No named author could be found on this webpage so the company abbreviation was instead used. The abbreviation has been expanded to aid clarity. Reviewer 2 pointed out “Line 783 on page 27, show as a subscript in the number of ‘Mo(O)2(acac)2’”. This has been revised. Reviewer 2 noted “Line 790-791 on page 27, delete the spaces in ‘5 - Substituted’, ‘2 - Nitrosophenol’, and ‘2 – Aroylbenzoxazoles”. This has been amended. Reviewer 2 pointed out “Line 797 on page 28, delete the spaces in ‘tri fl uoroacetic’. Line 807 on page 28, delete the spaces in ‘tri fl uoroacetic’”. This has been corrected. Reviewer 2 noted “Line 835 on page 28, delete the ‘<scp>’ and ‘</scp>’”. This has been resolved. Reviewer 2 noted “Line 840 on page 29, show as a subscript in the number of ‘Cu(Clqo)2’”. This has been amended Reviewer 2 pointed out “Line 893 and 898 on page 30, show as a subscript in the number of ‘CO2’”. This has been rectified.